# Synergistic Phototherapy-Molecular Targeted Therapy Combined with Tumor Exosome Nanoparticles for Oral Squamous Cell Carcinoma Treatment

**DOI:** 10.3390/pharmaceutics16010033

**Published:** 2023-12-26

**Authors:** Ming Li, Shiyao Yin, Anan Xu, Liyuan Kang, Ziqian Ma, Fan Liu, Tao Yang, Peng Sun, Yongan Tang

**Affiliations:** 1Jiangsu Key Laboratory of Neuropsychiatric Diseases, College of Pharmaceutical Sciences, Soochow University, Suzhou 215123, China; mli0919@suda.edu.cn (M.L.); 20214226024@stu.suda.edu.cn (A.X.); 20224226022@stu.suda.edu.cn (L.K.); 20235226014@stu.suda.edu.cn (Z.M.); liufan@suda.edu.cn (F.L.); tyang0920@suda.edu.cn (T.Y.); 2Department of Otolaryngology, the First Affiliated Hospital of Soochow University, Suzhou 215006, China; 20215232268@stu.suda.edu.cn

**Keywords:** nanomedicine, molecular targeted therapy, phototherapy, exosomes, oral squamous cell carcinoma

## Abstract

Oral squamous cell carcinoma (OSCC) contributes to more than 90% of all oral malignancies, yet the performance of traditional treatments is impeded by limited therapeutic effects and substantial side effects. In this work, we report a combinational treatment strategy based on tumor exosome-based nanoparticles co-formulating a photosensitizer (Indocyanine green) and a tyrosine kinase inhibitor (Gefitinib) (IG@EXOs) for boosting antitumor efficiency against OSCC through synergistic phototherapy-molecular targeted therapy. The IG@EXOs generate distinct photothermal/photodynamic effects through enhanced photothermal conversion efficiency and ROS generation, respectively. In vivo, the IG@EXOs efficiently accumulate in the tumor and penetrate deeply to the center of the tumor due to passive and homologous targeting. The phototherapy effects of IG@EXOs not only directly induce potent cancer cell damage but also promote the release and cytoplasmic translocation of Gefitinib for achieving significant inhibition of cell proliferation and tumor angiogenesis, eventually resulting in efficient tumor ablation and lymphatic metastasis inhibition through the synergistic phototherapy-molecular targeted therapy. We envision that the encouraging performances of IG@EXOs against cancer pave a new avenue for their future application in clinical OSCC treatment.

## 1. Introduction

Oral squamous cell carcinoma (OSCC), the most prevalent subsite of head and neck cancer, accounted for more than 90% of all oral malignancies that had new cases and associated deaths (377,713 and 177,757, respectively) in 2020 globally [1,2]. Though surgery, chemotherapy, and radiotherapy have been applied as the mainstay therapies for OSCC patients over the past two decades in clinics, the five-year survival rate remains no more than 50% [3,4]. Moreover, these therapy strategies invariably accompany substantial side effects, including functional damage and non-selective toxicity [5,6]. Recent progress in cancer genomics revealed that OSCC exclusively or over-expressed various surface receptors, such as epidermal growth factor receptor (EGFR), facilitating the application of molecular targeted therapy with improved selective toxicity. However, molecular targeted therapy drugs targeting EGFR, such as the hydrophobic molecule Gefitinib and chimeric monoclonal antibody cetuximab, suffered concerns of delivery difficulties and low response rates [7,8], leading to unremarkable clinical outcomes as demonstrated by the marginal improvements in patient median survival times [3,9]. Reasonably, applying combination therapy appears promising in amplifying antitumor performance through synergistic effects [10,11]. However, the efficient delivery of multiple drugs with various physicochemical features, pharmacokinetic profiles, and targeting behaviors to achieve synergistic combination therapy is still challenging [12,13].

Nanomedicines offer extraordinary platforms to tackle this challenge by delivering specific combinations and schedules of various therapeutics to the desired destinations of tumors due to the multifunctional feature and passive tumor targeting through the enhanced permeability and retention (EPR) mechanism, providing a chance to promote antitumor efficacy of combination therapy [14,15]. Cancer nanomedicines have achieved remarkable advancement over the past several decades as hundreds of nanomaterials have been developed as carriers [16]. Due to their distinct physicochemical properties compared with their bulkier counterparts, nanomaterials congenitally encounter concerns of biosafety and environmental explosion risk [15,17]. Encouragingly, after systematic safety evaluation on animals and humans, at least 15 cancer nanomedicines have been approved and are now widely used in clinics [18], indicating that the biosafety concerns of nanomaterials can be addressed, at least to a large degree, through rational design. Exosomes (EXOs) are extracellular vesicles originating from the late endosome with a membrane-bound structure and size range of ~40 to 160 nm [19]. Due to their intrinsic biocompatibility, rapid degradability, and low immunogenicity, bio-originated exosomes are widely deemed to be biofriendly with marginal environmental impact [20]. Moreover, the suitable dimension and abundant biomolecule on the surface not only endow exosomes with passive EPR effect but also intrinsic tissue-homing capabilities (homologous targeting) with minimal non-specific interactions, leading to dramatically enhanced tumor accumulation and penetration. Thus, these cell-derived vesicles opened new frontiers to construct biofriendly nanomedicines to deliver various therapeutic payloads for boosted cancer therapy [21,22].

Particularly, our previous works have demonstrated that photosensitizers containing nanomedicines can not only efficiently target tumors and exert a potent tumoricidal phototherapy effect (photodynamic or photothermal therapy) through photo-triggered reactive oxygen species (ROS) or hyperthermia, but also can promote intracellular translocation of antitumor drugs, leading to precise drug delivery and thus augmented anticancer outcomes [23,24,25,26]. Herein, we report that OSCC-derived exosomes co-formulating photosensitizer Indocyanine green (ICG) and EGFR inhibitor Gefitinib (IG@EXOs) boost the antitumor efficacy of OSCC through the synergistic phototherapy–molecular targeted therapy (PMTT) (Figure 1A). The IG@EXOs possess distinct photodynamic/photothermal effects, preferable cellular uptake, and effective tumor accumulation and penetration through the EPR effects and homologous targeting. Upon irradiation, the phototherapy effects of IG@EXOs not only directly induce cancer cell apoptosis but also simultaneously promote the release and translocation of Gefitinib to specifically bind with the intracellular target, finally resulting in potent antitumor performance through synergistic PMTT. The promising results of using IG@EXOs against OSCC open new pathways for its future application with other clinically relevant combinations.

## 2. Materials and Methods

### 2.1. Materials and Cells

ICG was obtained from Shanghai Yuanye Biological Technology Co., Ltd. (Shanghai, China). Gefitinib was purchased from Macklin Biochemical Co., Ltd. (Shanghai, China). RPMI 1640, DMEM, and MEM culturing media (Hyclone, UT, USA) were purchased from Cytiva Life Sciences (Marlborough, MA, USA). Lysotracker DND26 was purchased from Thermo Fisher Scientific Co., Ltd. (Waltham, MA, USA). Protease inhibitor cocktail, phosphatase inhibitor cocktail I, and CCK-8 were purchased from TargetMol Chemicals Inc. (Boston, MA, USA). Other agents and materials were obtained from Sinopharm Chemical Reagent Co., Ltd. (Shanghai, China). SCC7 oral squamous cell carcinoma cancer cells and human umbilical vein endothelial cells were obtained from the American Type Culture Collection (ATCC) and cultured in RPMI 1640 medium. 293T human embryonic kidney cells were obtained from ATCC and cultured in DMEM medium. FaDu human pharyngeal squamous cell carcinoma cell lines and L929 mouse fibroblast cells were purchased from IMMOCELL (Xiamen, China) and cultured in MEM medium. All cells were cultured with medium containing 10% FBS under standard conditions (37 °C, 5% CO_2_).

### 2.2. Preparation of Exosomes

SCC7 cancer cells were cultured in cell culture dishes with a diameter of 150 mm for 24 h. Then, the culture medium was collected for exosome extraction. After 30 min centrifugation at 20,000× *g* to remove dead cells and cell debris, blank SCC7 exosomes were obtained through further ultracentrifugation at 100,000 × *g* for 70.0 min. To prepare IG@EXOs, 2.0 mg of ICG and 2.0 mg of Gefitinib were mixed in 1.0 mL of deionized water, and 2.0 M HCl was used to adjust the solution pH value to 6.0. Then, the solution was added to the blank exosomes solution at the ratio of 1:4, followed by ultrasonication using an ultrasonic homogenizer (SCIENTZ-IID, Ningbo, China) at the power of 300 W for 15 min. The solution was further purified via dialysis (molecular weight cut-off of 8.0 kDa) against deionized water for 24 h to obtain IG@EXOs. The mixture of free ICG and Gefitinib (Free I/G) was prepared by dissolving the drugs in an aqueous solution (pH 6.0) containing 5.0% DMSO.

### 2.3. Characterization of Exosomes

The morphology of IG@EXOs was observed using a transmission electron microscope (TEM, Hitachi HT7700, Tokyo, Japan). The hydrodynamic diameters and zeta potential of IG@EXOs and blank EXOs were measured using a Zetasizer ZS90 (Malvern, Cambridge, UK). The absorption spectra of IG@EXOs and ICG were obtained using an ultraviolet-visible spectrophotometer (UV2600i, Shimadzu, Tokyo, Japan).

### 2.4. Western Blot

The proteins from the SCC7 cells, blank EXOs, and IG@EXOs were extracted and quantified by BCA protein assay. The proteins (20.0 μg) of each group were separated by a 10% SDS-polyacrylamide gel and then transferred to a nitrocellulose membrane, followed by blocking with a protein-free rapid blocking buffer. Then, the membrane was incubated with anti-Calnexin (CST), anti-TSG101 (CST), anti-CD9 (CST), and anti-CD63 (CST) antibodies at 4 °C for 24 h. Then, the nitrocellulose membrane was washed with TBST, followed by incubating with the secondary antibody (ZSGB-BIO) for 4 h at room temperature and imaged using an ECL-plus detection system (GE Healthcare, Chicago, IL, USA).

### 2.5. Photothermal Performance of IG@EXOs

A total of 0.5 mL of Free I/G or IG@EXOs (5.0 μg mL^−1^, ICG) was irradiated using a 785 nm light (0.5 W cm^−2^) for 5 min, and a thermometer was used to monitor the temperatures every 30 s. To investigate the photothermal conversion efficiency of IG@EXOs and Free I/G, 0.5 mL of solution (5.0 μg mL^−1^, ICG) was exposed to 785 nm light irradiation (0.5 W cm^−2^) for 5 min, followed by removing the light irradiation and cooling the solution down to room temperature, along with recording the solution temperature every 30 s. Then, the photothermal conversion efficiency was calculated by the following equation [27]:$$\eta =\frac{hA\left(T_{max}-T_{amb}\right)\;-\;Q_{0}}{I\left(1\;-\;10^{{-A}_{\lambda }}\right)}$$
where *h* is the heat transfer coefficient, *A* is the surface area of the container, *T_max_* is the maximum temperature during the process, *T_amb_* is the surrounding ambient temperature, *Q*_0_ is the heat input rate due to light absorption of the solvent, *I* is the light power, and *A_λ_* is the absorbance of the sample at an excitation wavelength of 785 nm.

### 2.6. Photostability and Chemical Stability

To determine the photostability, 5.0 mL of Free I/G or IG@EXOs (5.0 μg mL^−1^, of ICG) was irradiated by 785 nm light (0.5 W cm^−2^) for 0.5, 1, 2, 3, 5, 7, and 10 min. Then, the absorption spectra of the solution were measured at each time point. To measure the chemical stability, 5.0 mL of Free I/G or IG@EXOs (10.0 μg mL^−1^, ICG) was dispersed into various solutions including serum-free cell culture medium, ascetic acid ammonium acetate buffer (pH 5.0), and phosphate buffer (pH 7.4), followed by absorbance spectra monitoring at 0, 6, 12, 24, and 48 h.

### 2.7. Drug Release

To address the drug release behaviors, Gefitinib and ICG from Free I/G or IG@EXOs were evaluated using the dialysis method. A total of 1.0 mL of sample (0.2 mg mL^−1^, ICG) was packed in dialysis bags with a molecular weight cut-off of 3.5 kDa. Then, the drug-loaded dialysis bags were immersed into the ascetic acid ammonium acetate buffer (pH 5.0), followed by shaking with an oscillator shaker at 37 °C. The concentrations of ICG and Gefitinib in the buffer were measured at 0.5, 1, 2, 4, 6, 8, 12 and 24 h. To investigate the light-triggered drug release, the amounts of Gefitinib released from IG@EXOs were evaluated in ascetic acid ammonium acetate buffers (pH 5.0) after light irradiation (785 nm, 0.5 W cm^−2^) for 2, 5, and 10 min.

### 2.8. Quantum Yield of Singlet Oxygen (^1^O_2_)

To measure the ^1^O_2_ quantum yield (ΦΔ) of IG@EXOs, 1,3-diphenylbenzofuran (DPBF) was used as a probe, and free ICG was used as a reference (Φ_Δ_^ICG^ = 0.14). The solutions of samples (ICG and IG@EXOs) containing DPBF at 30.0 μM were exposed to a 785 nm light irradiation at 0.5 W cm^−2^ for 20 min, followed by recording the decrease in DPBF absorbance at 417 nm.

### 2.9. Cellular Uptake and Endocytic Pathway

SCC7 tumor cells (1.0 × 10^6^ cells/well) were seeded in six-well culture plates and incubated overnight at 37 °C. Then, Free I/G or IG@EXOs were added into the wells for 3, 6, 12, and 24 h incubation. ICG and Gefitinib in the cells were extracted using DMSO, followed by quantification through absorption and high-performance liquid chromatography (HPLC, 1100 Series, Agilent, Santa Clara, CA, USA) assay, respectively.

To investigate the endocytic pathway of IG@EXOs, cell culture mediums containing different inhibitors (10.0 μg mL^−1^ chlorpromazine, 100.0 μg mL^−1^ amiloride, and 50.0 μg mL^−1^ nystatin) were incubated with the cells for 6 h at 37 °C or 4 °C. Then, DiI-labelled IG@EXOs (10.0 μg mL^−1^, ICG) were added to the dish. After 2 h incubation, the cells were observed using confocal laser scanning microscopy (CLSM, Zeiss, LSM710, Oberkochen, Germany). DiI and Hoechst 33,342 channels were used for the detection of DiI-labelled IG@EXOs and nuclei, respectively. The imaging process was conducted in LineSequential mode with AxioObserver as the microscope and Plan-Apochromat 20×/0.8 M27 as the objective lens with laser power set to 2.0%, according to the reported methods with modifications [28,29]. The CLSM used below was operated under the same conditions.

### 2.10. Intracellular Distribution

The intracellular distribution of IG@EXOs was observed through CLSM using DiI-labeled IG@EXOs. SCC7 cells seeded in a glass bottom dish were incubated with DiI-labelled IG@EXOs for 3 h at 37 °C. Then, the cells were sequentially stained with 1.0 mL of Hoechst 33,342 (10.0 μg mL^−1^) for 10 min and 1.0 mL of Lysotracker Green DND-26 (50.0 nM) for 5 min, followed by CLSM observation. To visualize the subcellular distribution of IG@EXOs with light irradiation, cells pretreated with DiI-labelled IG@EXOs for 3 h were irradiated with a 785 nm light (0.5 W cm^−2^, 5 min), followed by incubation with Hoechst 33,342 and Lysotracker Green DND-26 and observation using CLSM (Zeiss, LSM710). DiI, Lysotracker Green DND-26, and Hoechst 33,342 channels were used for the detection of DiI-labelled IG@EXOs, lysosomes, and nuclei, respectively.

### 2.11. Dihydroethidium (DHE) and Acridine Orange (AO) Staining

To investigate the ROS generation of IG@EXOs in the cells, SCC7 cells pretreated with Free I/G or IG@EXOs (5.0 μg mL^−1^, ICG) were irradiated by a 785 nm light (5 min, 0.5 W cm^−2^), followed by DHE staining for 15 min and SYTO staining for 10 min at 37 °C. Then, the cells were observed using CLSM (Zeiss, LSM710).

Next, AO staining was performed to evaluate the disruption of lysosomal membranes. The SCC7 cells (5.0 × 10^4^ cells/well) pretreated with Free I/G or IG@EXOs (5.0 μg mL^−1^, ICG) were washed and incubated with fresh medium, followed by a 785 nm light irradiation of 5 min (0.5 W cm^−2^). Then, 1 h later, the cells were washed and incubated with AO solutions (6.0 μM, 1.0 mL) for 20 min. Then, the cells were washed and observed using CLSM (Zeiss, LSM710). Alexa Fluor 594 and SYTO channels were used for the detection of DHE and nuclei, respectively. The Alexa Fluor 514 channel was used for the detection of the acidic lysosome, and the Alexa Fluor 647 channel was used for the detection of neutralized cytosol and nucleus.

### 2.12. Cytotoxicity

SCC7 cells (5.0 × 10^3^ cells/well) were seeded into the 96-well plate and incubated at 37 °C overnight. Then, the cells were incubated with Free I/G or IG@EXOs at various concentrations (0, 0.5, 1.0, 2.0, 4.0, 8.0, 12.0, 16.0, 24.0, and 32.0 μg mL^−1^ of ICG) for 24 h. Then, the cells were irradiated by a 785 nm light (5 min, 0.5 W cm^−2^) or not. CCK-8 assay was used to evaluate the cell viability 24 h later.

Moreover, Calcein-AM/PI staining was further used to visualize the cytotoxicity. SCC7 cells were seeded on 24-well plates (5.0 × 10^5^ cells/well) and incubated overnight. Then, the cells were incubated with Free I/G or IG@EXOs (ICG, 10.0 μg mL^−1^) for 24 h. After that, the cells were washed and irradiated with a 785 nm light (5 min, 0.5 W cm^−2^), followed by incubating for another 24 h at 37 °C. Subsequently, the cells were stained by a Calcein-AM/PI kit and imaged through CLSM (Zeiss, LSM710). Calcein-AM and PI channels were used for the detection of live and dead cells, respectively.

### 2.13. 5-ethynyl-2′-deoxyuridine (EdU) Staining

To evaluate the influence of IG@EXOs on cell proliferation, SCC7 cells were seeded on 6-well plates (5.0 × 10^5^ cells/well) and incubated overnight. Afterward, Free I/G or IG@EXOs at 5.0 μg mL^−1^ of ICG were incubated with SCC7 cells for 24 h, followed by a 785 nm light irradiation (5 min, 0.5 W cm^−2^). Finally, the SCC7 cells were stained by EdU and observed by CLSM (Zeiss, LSM710). Alexa Fluor 488 and Hoechst 33,342 channels were used for the detection of proliferative cells and nuclei, respectively.

### 2.14. Apoptosis Assay

SCC7 cells were seeded on 6-well plates (5.0 × 10^5^ cells/well) and incubated overnight. Then, the cells were washed and incubated with Free I/G or IG@EXOs (5.0 μg mL^−1^, ICG) for another 24 h, followed by irradiation with a 785 nm light (5 min, 0.5 W cm^−2^). Then, 12 h later, the cells were stained with an Annexin V-FITC/PI apoptosis detection kit and analyzed using flow cytometry (Beckman Coulter, FC500, Brea, CA, USA).

### 2.15. Drug Penetration Studies in 3D SCC7 Tumor Cell Spheroids

SCC7 cells were seeded in 96-well round-bottom ultralow attachment standard plates (500 cells/well) and incubated for 5 days to establish tumor spheroids according to the literature method [30]. Then, the tumor spheroids with diameters of about 400 μm were incubated with DiI-labelled IG@EXOs or DiI-labelled IG@EXOs-FaDu (prepared using the human OSCC cells derived exosomes) (10 μg mL^−1^, ICG) for 6 h at 37 °C. Afterward, the tumor spheroids were washed, fixed, and transferred to glass-bottom dishes, followed by CLSM (Zeiss, LSM710) observation. The DiI channel was used for the detection of DiI-labelled IG@EXOs. Meanwhile, Free DiI was used as a control with the abovementioned procedures.

### 2.16. Establishment of Subcutaneous SCC7 Tumor Model

Female C3H mice at 6–8 weeks (18–20 g) were obtained from Beijing Vital River Laboratory Animal Technology Co., Ltd, Beijing, China. A total of 2.0 × 10^6^ SCC7 tumor cells (50.0 μL) were subcutaneously injected into the right back of the mouse. The tumor treatment experiments began when the tumor volume reached 70~100 mm^3^.

### 2.17. Ex Vivo Fluorescence Imaging and Biodistribution

For ex vivo fluorescence imaging, SCC7 tumor-bearing mice were administrated with IG@EXOs (ICG, 7.5 mg kg^−1^) via the tail vein. Then, 24 h later, various tissues, including the heart, liver, spleen, lung, kidney, and tumor, were extracted from the mice, followed by fluorescence imaging using an In Vivo Imaging Systems (IVIS, PerkinElmer, Lumina II, Hopkinton, MA, USA).

For biodistribution, methanol was used to extract ICG and Gefitinib from the heart, liver, spleen, lung, kidney, and tumor at 24 h post-injection. Finally, the amounts of ICG and Gefitinib were determined using an ultraviolet-visible spectrophotometer (UV2600, Shimadzu, Singapore) and HPLC (1100 Series, Agilent, Santa Clara, CA, USA), respectively. HPLC was performed according to the reported methods [31,32,33] with the following parameters: column, Cosmosil C18; mobile phase A, ammonium acetate solution; mobile phase B, acetonitrile; flow rate, 1.0 mL min^−1^; and temperature, 25 °C.

### 2.18. Tumor Penetration

DiI-labeled IG@EXOs (7.5 mg kg^−1^, ICG) were intravenously injected into the mice bearing SCC7 subcutaneous tumors. Then, the tumors were harvested and cut into 10.0 μm thick sections at 24 h post-injection, followed by fixing and blocking with 5.0% BSA. Afterward, the tumor sections were stained with anti-CD31 monoclonal antibody (1:200, Abcam, Cambridge, UK) and Alexa-Fluor 488 conjugated secondary antibody (1:100, Abcam) sequentially prior to observation under CLSM (Zeiss, Jena, Germany, LSM710). DiI, Alexa-Fluor 488, and DAPI channels were used for the detection of DiI-labelled IG@EXOs, blood vessels, and nuclei. To assess the penetration of IG@EXOs from the tumor blood vessels to the deep tumor tissues, the red fluorescence intensity of DiI-IG@EXOs was quantified along the direction vertically to the blood vessels.

### 2.19. In Vivo Infrared Thermography

Free I/G or IG@EXOs (7.5 mg kg^−1^, ICG) were intravenously injected into the SCC7 tumor-bearing female C3H mice. Then, the tumors were irradiated with a 785 nm light (5 min, 0.5 W cm^−2^) at 24 h post-injection, along with monitoring the temperature at the tumor region using an infrared camera (FLIR E50) during the irradiation.

### 2.20. In Vivo DHE Staining

Free I/G or IG@EXOs (7.5 mg kg^−1^, ICG) were intravenously injected into the SCC7 tumor-bearing female C3H mice. Then, the tumors were irradiated with a 785 nm light (5 min, 0.5 W cm^−2^) or not at 24 h post-injection. Next, the tumors were harvested and cut into 10.0 μm thick sections, followed by staining using DHE (10.0 μM) for 0.5 h and DAPI (10.0 μg mL^−1^) for 15 min. Finally, the tumor sections were observed using CLSM (Zeiss, LSM710). For details, the Alexa Fluor 594 channel was used for the detection of DHE.

### 2.21. In Vivo Antitumor Efficacy

Free I/G or IG@EXOs (ICG, 7.5 mg kg^−1^) were injected into the CC7 subcutaneous tumor-bearing C3H mice through the tail vein. Then, the tumors were irradiated with a 785 nm light or not (5 min, 0.5 W cm^−2^) 24 h post injection. Then, a vernier caliper (Deli, Ningbo, China, DL91150) was applied to measure the size of tumors during 11 days, and the volume was calculated by the equation of V = L × W^2^/2 (L represents the longest tumor size and W represents the shortest tumor size). At the end of the experiments, the tumors and lymph nodes of the mice were harvested for weighting.

### 2.22. Histological Staining and Serum Biochemistry

To further investigate the cell damage induced by the synergistic PMTT, SCC7 subcutaneous tumor-bearing C3H mice were intravenously injected with Free I/G or IG@EXOs (7.5 mg kg^−1^, ICG), followed by a 785 nm light irradiation at 0.5 W cm^−2^ for 5 min or not at 24 h post-irradiation. Another 6 h later, tissues including the heart, liver, spleen, lung, spleen, and tumor were extracted and cut into 10 μm thin sections for hematoxylin-eosin (H&E) staining. Meanwhile, the tumors were further cut into other sections for Ki67 and TUNEL staining. Lymph nodes harvested at the end of the antitumor experiment were stored in the 4.0% formaldehyde solution and cut into thin sections for H&E staining. Finally, H&E or Ki67 stained sections were observed using an IX73 bright field microscope (Olympus, Tokyo, Japan), and the TUNEL stained sections were observed using CLSM (Zeiss, LSM710). Alexa Fluor 488, Alexa Fluor 647, and DAPI channels were used for the detection of TUNEL-positive cells, Ki67-positive cells, and nuclei, respectively.

To investigate the biosafety of IG@EXOs, healthy C3H mice were intravenously injected with Free I/G or IG@EXOs (7.5 mg kg^−1^, ICG). Then, the liver and kidney function markers, including alkaline phosphatase (ALP), alanine transaminase (ALT), aspartate aminotransferase (AST), and urea were detected at 0, 7, 14, 21, and 28 days post-injection. Moreover, various routine blood parameters such as red blood cell count (RBC), white blood cell count (WBC), and platelet count (PLT) of mice treated with IG@EXOs were evaluated at 3 days post-injection.

### 2.23. Statistical Analysis

All data were expressed as mean ± s.d. The differences between groups were assessed using a two-sided Student’s *t*-test or one-way ANOVA with Tukey’s post hoc test. Statistical differences were defined as ** p* < 0.05, *** p* < 0.01, and **** p* < 0.001.

## 3. Results and Discussions

### 3.1. Preparation and Characterization of IG@EXOs

IG@EXOs co-formulating ICG and Gefitinib were prepared with ultracentrifugation and sonication procedures. The TEM images showed that optimized IG@EXOs with ICG and Gefitinib loading rates of 11.3% and 9.2%, respectively, displayed spherical morphology with an average diameter of 62.3 ± 4.7 nm (Figure 1B). Dynamic light scattering shows that the IG@EXOs had an average hydrodynamic diameter of 83.7 nm, a polydispersity index of 0.215 (Figure 1C), and zeta potential of 15.4 mV (Figure 1D), which were similar to the blank exosomes (Appendix A), indicating the drug loading process has marginal influence on the exosome structure. The suitable size, monodispersity, and negative surface potential of IG@EXOs are highly beneficial for tumor targeting and internalization [22,34,35]. Western blot analysis exhibited a significant level of marker proteins like TSG101, CD63, and CD9 (Figure 1E), verifying the exosome nature of the nanoparticles. The IG@EXOs displayed an enhanced absorption peak at 800 nm, which was red-shifted from the original peak of free ICG at 778 nm (Figure 1F), indicating the J-type aggregation of ICG molecules through intermolecular π-π stacking within the exosomes [36,37]. IG@EXOs showed enhanced photostability in aqueous conditions (Appendix A), reasonably because of the encapsulation and interaction of the cargoes within the nanocarriers.

ICG is a clinically approved photosensitizer with apparent photothermal and photodynamic effects [38,39,40]. Thus, we subsequently evaluated the photothermal effects of IG@EXOs. Remarkably, the IG@EXOs at 10.0 μg mL^−1^ of ICG with light irradiation (785 nm, 0.5 W cm^−2^) for 5 min displayed a potent temperature increase of 19.4 °C, being much higher than that of free ICG (~13.7 °C) (Figure 1G), which might be ascribed to the enhanced non-radiative transition of the ICG aggregation within the exosomes [36]. The photothermal conversion efficiency of IG@EXOs under 785 nm light irradiation was further calculated to be 27.6%, being higher than that of free ICG (~21.4%) (Appendix A), which reasonably evidenced the potent photothermal effects of IG@EXOs. These results verify that IG@EXOs can amplify the photothermal effect of photosensitizer, which is highly advantageous for hyperthermia-mediated phototherapy against malignant tumors.

Then, we investigated the photodynamic effects of IG@EXOs using a singlet oxygen sensor green (SOSG) as the probe. As shown in Figure 1H, IG@EXOs under irradiation exhibited drastically increased fluorescence intensity, indicating the generation of ROS. Next, we measured the ^1^O_2_ quantum yield using DPBF as a probe. Distinctly, IG@EXOs showed a ^1^O_2_ quantum yield of 0.23, being much higher than that of the free ICG (~0.14) (Appendix A), which might be due to the microenvironment inside the nanocarrier enhanced singlet-to-triplet transition and subsequent energy transfer to molecular oxygen [41,42].

To investigate the drug release profiles, IG@EXOs samples exposed to light irradiation for various minutes were sealed in dialysis bags and immersed in ascetic acid ammonium acetate buffer with a pH value of 5.0, followed by quantifying the free Gefitinib in the solution for 24 h. As shown in Figure 1I, IG@EXOs displayed sustained releases of Gefitinib with an accumulative release ratio of 57.8% in the acid buffer without light irradiation, indicating the moderate drug release in the acid lysosome of tumor cells. In contrast, light irradiation obviously elevated Gefitinib release ratios up to 62.8%, 74.1%, and 92.9% for 2, 5, and 10 min irradiation, respectively (Figure 1I). The stimuli response drug release might be due to the photoinduced hyperthermia/ROS that destructed the exosome structure, as evidenced by the TEM images (Appendix A), suggesting the superiority of IG@EXOs as intelligent platforms for cancer therapeutics delivery. Moreover, IG@EXOs showed satisfactory stability in aqueous solutions (Appendix A), as further proved by the ignorable variations of hydrodynamic size and zeta potential during 15 days of storage at 4 °C (Figure 1J,K).

### 3.2. Cellular Behaviors of IG@EXOs

To investigate the cell uptakes, the cellular concentrations of ICG and Gefitinib were assessed after incubating IG@EXOs or Free I/G with SCC7 tumor cells. Figure 2A,B showed that SCC7 tumor cells exhibited significant internalization of ICG and Gefitinib in a time-dependent manner after incubation with IG@EXOs or Free I/G. Notably, IG@EXOs displayed a 1.9–2.4-fold and 1.6–1.9-fold increment in internalized ICG and Gefitinib, respectively, compared with Free I/G, which is highly advantageous for the efficient delivery of ICG and Gefitinib to their intracellular targets. Subsequently, the endocytic pathway of IG@EXOs was verified using different endocytic inhibitors. SCC7 tumor cells displayed a 53.4%, 57.3%, and 59.3% decrease in internalized IG@EXOs upon treatment with chlorpromazine, amiloride, and 4 °C, respectively, indicating energy-dependent clathrin-mediated pathway and macropinocytosis of IG@EXOs (Figure 2C,D) [23,35,43,44,45]. Then, CLSM was used to investigate the intracellular behaviors of the internalized IG@EXOs (Figure 2E,F). After 2 h incubation with SCC7 cells, IG@EXOs showed a high co-localization of 95.2% with lysosomes. However, after exposure to light irradiation at 0.5 W cm^−2^ for 5 min, the IG@EXOs/lysosome co-localization rate blatantly dropped to 28.3%, indicating that the light irradiation facilitates the lysosomal membrane rupture and lysosome-cytoplasm translocation of IG@EXOs. The lysosomal rupture was further demonstrated by the AO staining experiment. AO is an intracellular probe that shows red fluorescence in acidic lysosomes and emits green fluorescence in neutralized cytosol and nuclei [41,46]. The AO displayed red fluorescence in cells treated with either IG@EXOs or irradiation. Meanwhile, cells treated with Free I/GG under light irradiation also displayed evident red fluorescence of AO, indicating the presence of intact lysosomes after these treatments (Figure 2G). In contrast, IG@EXOs with light irradiation resulted in the disappearance of red fluorescence (Figure 2G), indicating evident lysosomal destruction. The lysosomal destruction might be ascribed to the light-induced ROS generation from IG@EXOs, which was further validated by the enhanced red fluorescence from the ROS probe DHE, as shown in Figure 2H,I. Thus, IG@EXOs exhibited significant light-induced ROS, reasonably resulting in the lysosomal membrane rupture in a relatively short period to facilitate the cytoplasmic translocation of Gefitinib for inducing cell apoptosis.

### 3.3. Cytotoxicity of IG@EXOs

The cytotoxicity of IG@EXOs and Free I/G with or without light irradiation (785 nm, 0.5 W cm^−2^, 5 min) was assayed using SCC7 murine OSCC tumor cells (Figure 3A). Free I/G exhibited relatively weak cytotoxicity with IC_50_ of 24.3 and 34.2 μg mL^−1^ of ICG with and without light irradiation, respectively. In contrast, IG@EXOs without irradiation exhibited enhanced cytotoxicity with IC_50_ of 14.1 μg mL^−1^ of ICG, possibly due to the enhanced internalization of Gefitinib by the exosomes. Notably, IG@EXOs with light irradiation showed a dramatically increased cytotoxicity with IC_50_ of 7.4 μg mL^−1^ of ICG, indicating the vital contributions of light activation on cell killing. To evaluate the biosafety of IG@EXOs, we tested the cytotoxicity of IG@EXOs against normal cell lines, including the L929 mouse fibroblast cell line, 293T human embryonic kidney cells, and human umbilical vein endothelial cells. The results showed that these cells retain high viability even under high concentrations of IG@EXOs (32 μg mL^−1^), indicating the good biosafety of IG@EXOs (Appendix A). Meanwhile, we also conducted Calcein-AM/PI staining to investigate the cytotoxicity of IG@EXOs and Free I/G. As shown in Figure 3B,C, PBS showed no cytotoxicity on SCC7 cells regardless of light irradiation, and the Free I/G exhibited fragile cell killing ability, with more than 80.1% of SCC7 cells remaining alive. However, upon exposure to 785 nm light irradiation, only 23.0% of SCC7 cells treated with IG@EXOs were still alive, indicating that IG@EXOs possess potent photo-activated cytotoxicity.

Next, to further evaluate the ability of IG@EXOs to inhibit cell proliferation, EDU staining of SCC7 cells treated with IG@EXOs or Free I/G was carried out with and without light irradiation (785 nm, 0.5 W cm^−2^, 5 min). As shown in Figure 3D,E, PBS regardless of light irradiation and Free I/G alone showed nearly no influence on cell proliferation. Meanwhile, Free I/G under 785 nm light irradiation and IG@EXOs alone showed cell proliferation reductions of 33.8% and 37.3%, respectively, implying moderate cytotoxicity of photoactivated Free I/G and IG@EXOs in the dark. In contrast, IG@EXOs with light irradiation showed a dramatically decreased proliferation of SCC7 cells, with only 17.4% of cells maintaining proliferation ability, indicating the vital contributions of light on the cytotoxicity of IG@EXOs.

To investigate the mechanism of cell damage induced by photoactivated IG@EXOs, flow cytometry was applied to monitor the apoptotic levels using Annexin V-FITC/PI staining. Figure 3F,G show that PBS-treated tumor cells exhibited no evident apoptosis despite light irradiation. Meanwhile, Free I/G under irradiation exhibited mild apoptosis with early and late apoptosis levels of 18.7% and 17.3%, respectively, which might be ascribed to the combination of phototherapy originating from ICG and molecular targeted therapy through Gefitinib. In contrast, IG@EXOs under irradiation revealed a distinctly improved apoptosis with early and late apoptosis levels of 37.2% and 29.8%, respectively, significantly higher than that of IG@EXOs without irradiation (15.1% and 8.6% of early and late apoptosis, respectively). The promoted apoptotic levels by photoactivated IG@EXOs might be attributed to the collective effects of enhanced cell uptake and synergistic PMTT.

The apoptotic mechanism of IG@EXOs was further assessed by measuring the level of marker protein in SCC7 cells receiving various treatments by immunofluorescence staining. Tumor cells treated with IG@EXOs under light irradiation displayed the highest level of Cl-Caspase-3, a key protein in the execution phase of cell apoptosis, compared to other control groups (Figure 3H). Flow cytometric analysis further validated the promoted Cl-Caspase-3 level in cells receiving photoactivated IG@EXOs treatment (Figure 3I), implying the vital contribution of cell apoptosis on the photocytotoxicity of IG@EXOs.

### 3.4. Tumor Targeting and Penetrating of IG@EXOs

To evaluate the in vivo biodistribution behavior, IG@EXOs or Free I/G at 7.5 mg kg^−1^ of ICG were intravenously injected into the mice bearing SCC7 tumors, followed by ex vivo fluorescence imaging of the major organs and tumors at 24 h post-injection through detecting the fluorescence of ICG. As shown in Figure 4A,B, IG@EXOs displayed a four-fold increase in ICG fluorescence of tumors as compared to Free I/G, indicating efficient tumor targeting of IG@EXOs, probably through the EPR effect and homologous targeting. We further verified the biodistribution of IG@EXOs by quantifying ICG and Gefitinib in the major organs and tumors. Figure 4C,D showed ICG and Gefitinib mainly distributed into the liver, spleen, and tumor, with tumor distribution of ~6.9% ID g^−1^ and ~7.2% ID g^−1^ for ICG and Gefitinib, respectively. Reasonably, the preferable tumor accumulation of IG@EXOs would facilitate the synergistic PMTT against OSCC tumors due to the concentration-dependent cytotoxicity.

Delivering drugs into the deep tumor region is of vital importance for boosting antitumor efficacy [47,48,49,50]. To exploit the penetrability of IG@EXOs in tumors, mice bearing SCC7 subcutaneous tumors were intravenously injected with IG@EXOs at 7.5 mg kg^−1^ of ICG, followed by tumor section at 24 h post-injection and immunofluorescence imaging to observe the distribution of IG@EXOs. IG@EXOs showed significant extravasation from the blood and distributed uniformly in the tumor tissue (Figure 4E,F), indicating good penetrability of IG@EXOs. The deep penetration of IG@EXOs in tumors should reasonably promote killing cancer cells in both the peripheral and central tumor regions, resulting in amplified antitumorigenic outcomes. We further verified the penetrability of IG@EXOs by monitoring the penetration behaviors of IG@EXOs in SCC7 3D tumor spheres through CLSM (Zeiss, LSM710) imaging. The results showed that IG@EXOs exhibited a uniform distribution throughout all the SCC7 3D tumor spheres, and strong fluorescence from DiI-labelled IG@EXOs could be detected at a depth of 80 μm. In contrast, the control group of Free DiI only showed fluorescence from the peripheral regions, with nearly no fluorescence signal in the central part (80 μm depth) (Figure 4G,H). Interestingly, IG@EXOs-FaDu, which was prepared using exosomes derived from human OSCC cells, showed decreased penetration ability in SCC7 3D tumor spheres, as evidenced by the moderate fluorescence at a depth of 60 μm and slight fluorescence at a depth of 80 μm (Figure 4G,H). These data indicate that homologous targeting contributes greatly to the deep penetration of IG@EXOs into SCC7 tumors. The preferable tumor targeting and penetration capability of IG@EXOs offers significant advantages for potentiating the antitumor efficacy.

### 3.5. In Vivo Photo-Induced Hyperthermia and ROS and Antiangiogenesis Performance

To investigate the in vivo photothermal effect, IG@EXOs and Free I/G (7.5 mg kg^−1^ of ICG) were intravenously injected into the SCC7 tumor-bearing mice, followed by 5 min irradiation (785 nm, 0.5 W cm^−2^) at 24 h post-injection while the tumor temperature elevations (ΔT) were monitored. Both IG@EXOs and Free I/G exhibited time-dependent temperature elevations at the tumor site, verifying apparent photothermal effects. Particularly, PBS and Free I/G only induced modest (ΔT = 2.7 °C) and mild hyperthermia (ΔT = 11.3 °C) under irradiation for 5 min, respectively, while IG@EXOs dramatically raised the tumor temperature by ~15.0 °C under the same condition (Figure 5A,B), which might be attributed to the enhanced tumor accumulation and elevated photothermal conversion of IG@EXOs. The preferable photothermal effects of IG@EXOs facilitate direct cancer cell damage through photo-induced hyperthermia (>45 °C) [27].

To demonstrate the in vivo photodynamic effect, IG@EXOs and Free I/G (7.5 mg kg^−1^ of ICG) were intravenously injected into the SCC7 tumor-bearing mice, followed by 5 min irradiation of the tumors (785 nm, 0.5 W cm^−2^) at 24 h post-injection. Then, the tumor sections were prepared and stained with a ROS probe, DHE, which produces red fluorescence by transforming into 2-hydroxyehtidium after encountering ROS. As shown in Figure 5C,D, photoactivated IG@EXOs treatment induced intense red fluorescence of the tumor tissue, which was more than three-fold compared to that of Free I/G under the same conditions, verifying the potent ROS generation capability of IG@EXOs, due to enhanced tumor accumulation and an elevated ^1^O_2_ quantum yield of IG@EXOs. Moreover, after treatment with a ROS scavenger, vitamin C, the red fluorescence of the tumor tissues significantly decreased, further validating the abundant ROS generation of IG@EXOs under irradiation. ROS is a highly reactive component that can not only directly induce cancer cell apoptosis by destroying cell structures but also break the exosome membrane by oxidating the lipid, promoting drug release [51,52].

As a selective EGFR-TK inhibitor, Gefitinib can bind the intracellular domain of the EGFR receptor to block the corresponding signal transduction, preventing VEGFR-dependent angiogenesis [53,54]. To assay the antiangiogenesis ability, IG@EXOs and Free I/G (7.5 mg kg^−1^ of ICG) were intravenously injected into the SCC7 tumor-bearing mice, followed by exposing the tumors to light irradiation (785 nm, 0.5 W cm^−2^, 5 min) at 24 h post-injection. Then, the tumor sections were prepared and stained with an anti-CD31 antibody and Goat Anti-Rabbit IgG H&L (Alexa Fluor^®^ 488) secondary antibody (pseudo-colored green) on the third-day post-irradiation, followed by assaying the vascular morphology through CLSM observation. As shown in Figure 5E, the dramatically decreased green fluorescence from the mice treated with IG@EXOs upon light irradiation indicated a reduction in the quantity and length of tumor blood vessels, which was further verified by the statistics, showing a 31.4% decrease compared to PBS control (Figure 5F). The shrinkage of tumor blood vessels could reasonably lead to the inhibition of OSCC tumors. Collectively, IG@EXOs exhibited a potent photothermal/dynamic effect under irradiation, which can not only directly induce cytotoxicity but also promote the precise delivery of Gefitinib to the intracellular target, reasonably leading to boosted antitumor performance through synergistic PMMT.

### 3.6. Antitumor Efficacy against SCC7 Tumor

To elucidate the tumor suppressive efficacy, IG@EXOs and Free I/G at 7.5 mg kg^−1^ of ICG were intravenously administrated into the mice bearing subcutaneous SCC7 tumor models (average tumor volume of ~100 mm^3^), followed by 5 min irradiation (785 nm, 0.5 W cm^−2^) or not at 24 h post-injection. Subsequently, the tumor volumes were measured for 11 days. Figure 6A,B show that PBS-treated mice revealed rapid tumor growth regardless of irradiation, suggesting that light irradiation alone does not impact tumor growth. Free I/G with and without light irradiation exhibited moderate tumor inhibition rates of 34.6% and 47.2%, respectively, and IG@EXOs without light irradiation exhibited tumor inhibition rates of 17.6%. These results indicate that the synergy of ICG/Gefitinib in a free form or IG@EXOs alone is not potent enough to generate sufficient anticancer efficacy. Remarkably, IG@EXOs with light irradiation efficiently suppressed tumor growth with a tumor inhibition rate of 88.6%, revealing the preferable antitumor efficacy achieved by potently synergistic PMTT. At the end of the experiment, the mice were sacrificed to assay the tumor weights. PBS despite irradiation, Free I/G despite irradiation, and IG@EXOs without light irradiation showed insignificant or modest tumor suppression efficacy with tumor weight of about 58.8–97.8% to the PBS group, while IG@EXOs with irradiation showed dramatically decreased tumor weight (17.1% compared to the PBS group) (Figure 6C), reiterating the preferable antitumor performance of photoactivated IG@EXOs through synergistic PMTT.

Lymphatic metastasis is one of the major challenges in treating OSCC, contributing to the majority of patient deaths [12,55]. To evaluate whether the PMTT of IG@EXOs can inhibit the lymphatic metastasis of OSCC, we further assayed the lymph nodes at the end of the antitumor experiment. PBS despite irradiation, Free I/G despite irradiation, and IG@EXOs without light irradiation showed much larger lymph nodes with a weight of 79.3–128.7 mg and volume of 117.6–208.8 mm^3^ (Figure 6D,E and Appendix A). In contrast, mice treated with IG@EXOs under irradiation exhibited the smallest lymph nodes with a weight of 1.7 mg and volume of 5.6 mm^3^ (Figure 6D,E and Appendix A), indicating the photoactivated IG@EXOs can dramatically inhibit the lymphatic metastasis of SCC7 tumor, which may be due to the potent PMTT effects. Moreover, the H&E staining results further validated those mice of the control groups displayed evident metastatic nodules. In contrast, lymph nodes from the mice receiving IG@EXOs with light irradiation treatments have no tumor metastasis (Figure 6F), further verifying metastasis inhibition by IG@EXOs through the synergistic PMTT, which is highly advantageous to improve the prognosis of OSCC.

To further verify the cancer cell damage induced by the synergistic PMTT, tumors of mice suffering from IG@EXOs or Free I/G treatment in the presence or absence of light irradiation were harvested for H&E staining assay. IG@EXOs with light irradiation led to severe tumor necrosis and intense hemorrhagic inflammation in tumor tissues, while PBS and IG@ EXOs without irradiation had no apparent influence on tumor cells (Figure 6G). The Ki67 and TUNEL staining also verified that mice treated with IG@EXOs under light irradiation led to the most tumor cell apoptosis and the least Ki67-positive cancer cell with proliferation ability (Figure 6G). These data suggested that IG@EXOs inhibit tumor growth by inducing cancer cell apoptosis under irradiation.

Moreover, the body weights of mice with various treatments exhibited no obvious difference, suggesting good biocompatibility of IG@EXOs (Appendix A). Then, we tentatively evaluate the biosafety of the IG@EXOs through serum biochemistry assay and routine blood tests of healthy mice receiving intravenous injections of IG@EXOs or Free I/G. Figure 7A–D showed that the levels of liver function markers like ALP, ALT, and AST, and kidney function marker (urea) of mice treated by IG@EXOs or Free I/G exhibited insignificant changes compared to the PBS group, suggesting IG@EXOs have no remarkable toxicity on liver and kidney functions. Moreover, various routine blood parameters such as RBC, WBC, and PLT of mice treated with IG@EXOs showed no significant difference compared with healthy untreated mice (Figure 7E–G), indicating that IG@EXOs did not induce abnormal immune or inflammatory responses in the mice. In addition, IG@EXOs displayed no significant injury on the normal tissues, including the heart, kidney, spleen, lung, and liver, as evidenced by the H&E staining results (Figure 7H), demonstrating the negligible dark-cytotoxicity and distinct biosafety of IG@EXOs. These data collectively suggested the good biosafety of IG@EXOs.

ICG and Gefitinib are FDA-approved pharmaceuticals for human use, ensuring their superior biosafety. Moreover, exosomes, as bioinspired nanocarriers, exhibit intrinsic biocompatibility, biodegradability, and low immunogenicity as compared with artificially synthesized organic polymers and inorganic nanoparticles [56]. In fact, at least six exosome-based therapeutics have proceeded into phase I/II clinical trials for drug delivery [57]. Taken together, both the experiment results and the published literature suggest that IG@EXOs possess satisfactory biosafety. In addition, OSCC-derived exosomes as carriers exhibit elevated tumor accumulation through the passive EPR effects and active homologous targeting, which not only enhances the antitumor efficacy but also diminishes the casual damage to the normal tissue by decreasing the off-target distribution of drugs. The synergistic PMMT strategy may lead to adequate tumor outcomes at lower administration doses, further relieving the biosafety concerns. However, the application of tumor exosome-based nanoparticles in the real world is still largely impeded by several scientific issues and engineering challenges, such as low drug loading efficiency, the unclear fate of exosomes in the body, lack of a large-scale manufacturing strategy conforming to good manufacturing practice (GMP), and difficulty in quality control [57].

## 4. Conclusions

We have developed a tumor exosome-based nanoparticle co-formulating ICG/Gefitinib (IG@EXOs) for synergistic PMTT against oral squamous cell carcinoma. IG@EXOs possessed uniform dimensions and exosome surface markers, along with the efficient encapsulation of ICG as a photosensitizer and Gefitinib as a molecular targeted drug. IG@EXOs could sensitively respond to light irradiation with evident generation of hyperthermia and ROS, which not only directly induced cancer cell apoptosis but also triggered the on-demand release and cytoplasmic translocation of Gefitinib, resulting in the synergistic PMTT. Moreover, IG@EXOs could efficiently target and penetrate the OSCC tumor through both the EPR effects and homologous targeting, leading to the high-concentration accumulation of IG@EXOs at the tumor region. The synergistic PMTT and the efficient tumor targeting of IG@EXOs finally led to dramatic tumor elimination and lymphatic metastasis inhibition. The boosted therapy efficacy against OSCC by this exosome-based nanoparticle provides new clues for designing combinational treatments for cancer therapy.

## Figures and Tables

**Figure 1 pharmaceutics-16-00033-f001:**
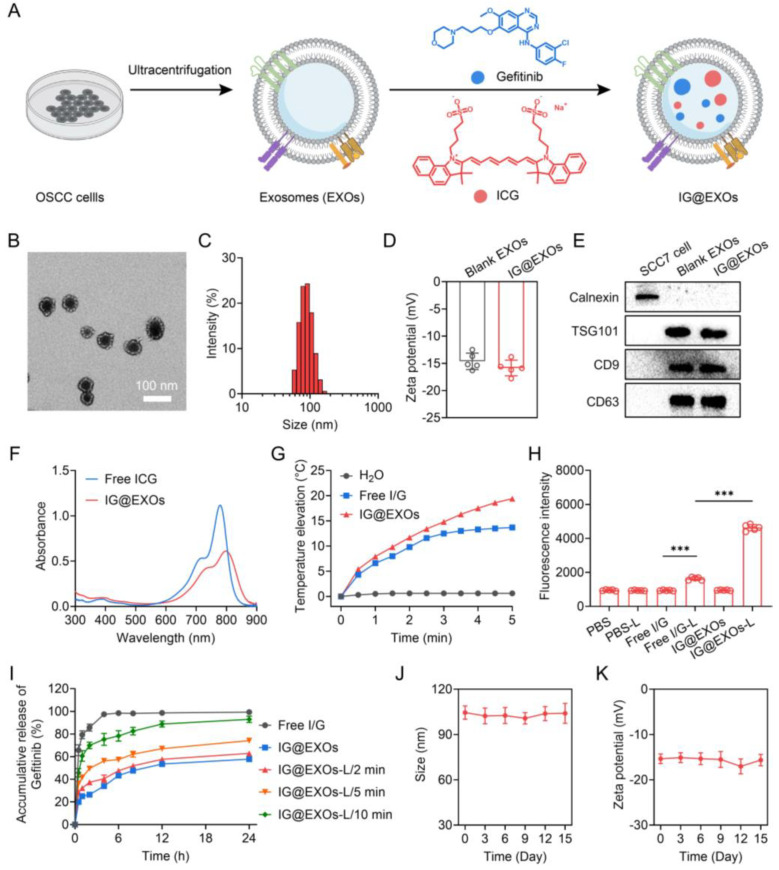
Scheme illustration and characterization of IG@EXOs. (**A**) Scheme illustrating the preparation procedure of IG@EXOs. TEM image (**B**) and hydrodynamic size distribution (**C**) of IG@EXOs. (**D**) Zeta potential of blank EXOs and IG@EXOs. (**E**) Western blot analysis of Calnexin, TSG101, CD9, and CD63 of IG@EXOs. (**F**) Absorption spectra of IG@EXOs and free ICG. (**G**) Temperature elevations of IG@EXOs under 785 nm light irradiation (0.5 W cm^−2^). (**H**) SOSG fluorescence intensity of various samples with light irradiation or not (785 nm, 0.5 W cm^−2^). (Statistical differences were defined as *** *p* < 0.001). (**I**) Accumulative releases of Gefitinib from Free I/G or IG@EXOs with light irradiation (785 nm, 0.5 W cm^−2^) for various minutes. Changes of hydrodynamic size (**J**) and zeta potential (**K**) of IG@EXOs stored at 4 °C for 15 days.

**Figure 2 pharmaceutics-16-00033-f002:**
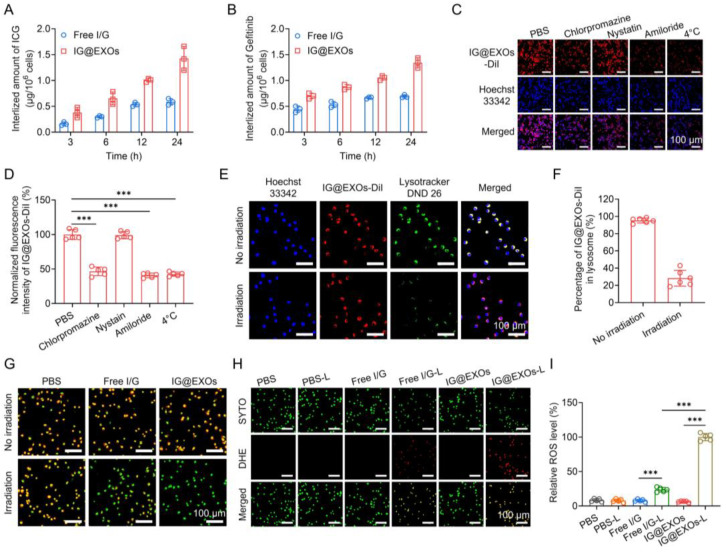
Cellular behaviors of IG@EXOs. Amount of internalized ICG (**A**) and Gefitinib (**B**) of Free I/G or IG@EXOs by SCC7 cells at different time points. Confocal images of SCC7 cells incubated with IG@EXOs-DiI and various endocytic pathway inhibitors (**C**) and the corresponding fluorescence intensities (**D**). Confocal images show the co-localization of IG@EXOs and lysosomes under light or not (**E**) and the corresponding statistical analysis (**F**). (**G**) AO staining fluorescence images of cells under 785 nm light irradiation or not. DHE staining images of cells under 785 nm light irradiation or not (**H**) and the corresponding fluorescence intensities analysis (**I**). Statistical differences were defined as *** *p* < 0.001 in Figure 2D,I.

**Figure 3 pharmaceutics-16-00033-f003:**
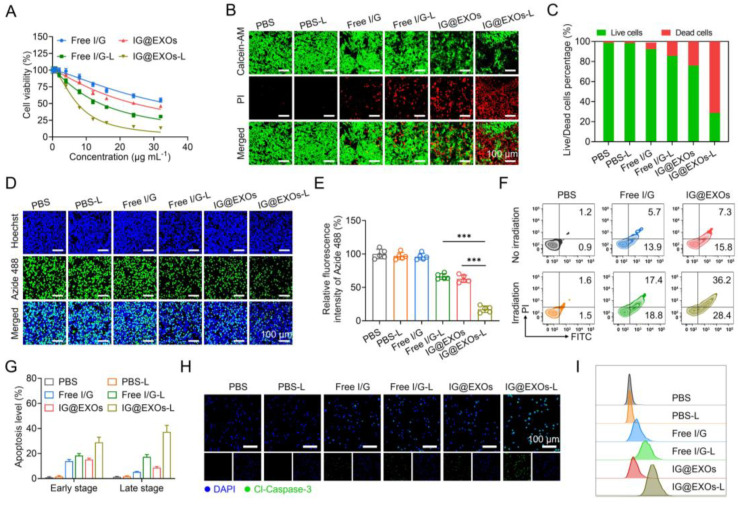
Cytotoxicity of IG@EXOs. (**A**) Cell viability of SCC7 cells treated with Free I/G or IG@EXOs under 785 nm light irradiation at 0.5 W cm^−2^ for 5 min. Calcein-AM/PI fluorescence images of SCC7 tumor cells after different treatments (**B**) and the corresponding quantitative statistical analysis of live/dead cells (**C**). Cell proliferation analysis by EdU staining (**D**) and the corresponding statistical analysis (**E**). Statistical differences were defined as *** *p* < 0.001. Cell apoptosis analysis with flow cytometry of SCC7 tumor cells after different treatments (**F**) and the corresponding statistical analysis (**G**). Immunofluorescence staining (**H**) and flow cytometry plot (**I**) of Cl-Caspase-3 of SCC7 tumor cells after different treatments.

**Figure 4 pharmaceutics-16-00033-f004:**
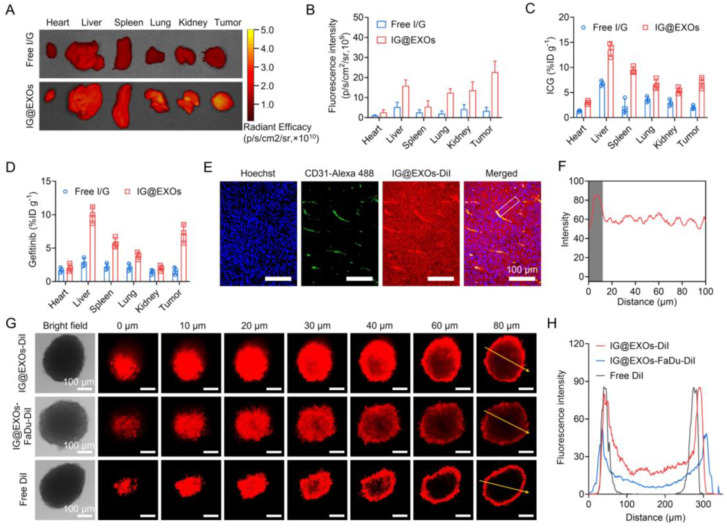
Tumor targeting and penetration abilities of IG@EXOs. Ex vivo fluorescence images of various tissues at 24 h post-injection of Free I/G or IG@EXOs (ICG, 7.5 mg kg^−1^) (**A**) and the corresponding averaged fluorescence intensity (**B**). Biodistribution of ICG (**C**) and Gefitinib (**D**) at 24 h post-injection of Free I/G or IG@EXOs. Immunofluorescence imaging of IG@EXOs-DiI (red) and blood vessels (green) in the SCC7 tumor sections at 24 h post-injection (**E**) and the corresponding fluorescence intensities of IG@EXOs-DiI quantified along the long dimension of the white box (**F**). Penetration efficiency of IG@EXOs in SCC7 3D tumor spheres (**G**) and the corresponding fluorescence intensities analysis (**H**).

**Figure 5 pharmaceutics-16-00033-f005:**
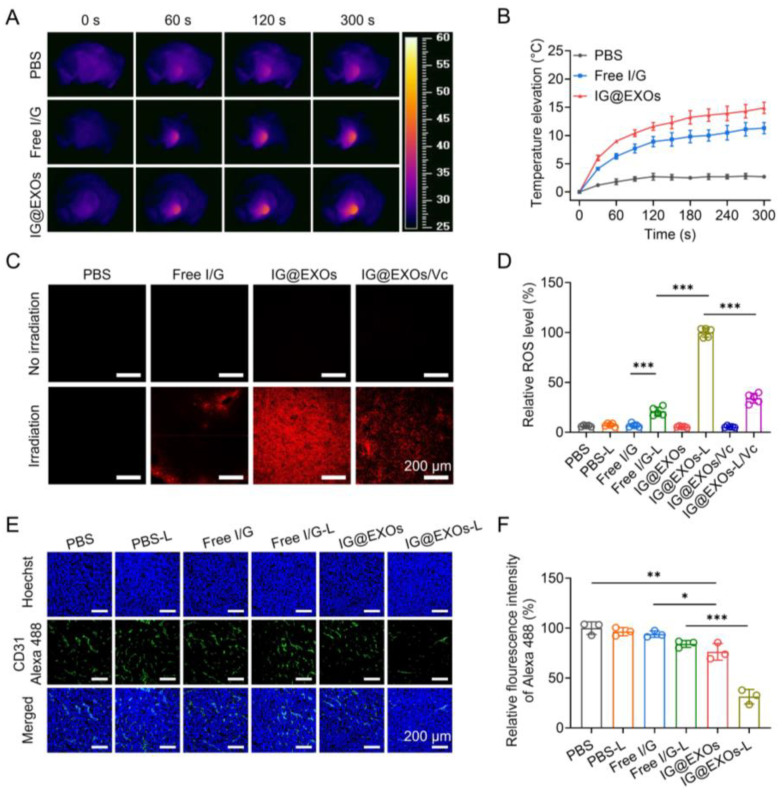
In vivo photothermal/dynamic effects and antiangiogenesis performance of IG@EXOs. Infrared thermography (**A**) and tumor temperature elevation curves (**B**) of SCC7 subcutaneous tumor-bearing mice treated with Free I/G or IG@EXOs under irradiation. DHE staining images of tumors harvested from SCC7 subcutaneous tumor-bearing mice treated with IG@EXOs at 24 h post-injection under 785 nm irradiation (0.5 W cm^−2^, 5 min) (**C**) and the corresponding statistical analysis (**D**). Immunofluorescence imaging of tumor blood vessels of SCC7 subcutaneous tumor-bearing mice receiving different treatments (**E**) and the corresponding fluorescence analysis (**F**). Statistical differences were defined as * *p* < 0.05, ** *p* < 0.01, and *** *p* < 0.001 in Figure 5D,F.

**Figure 6 pharmaceutics-16-00033-f006:**
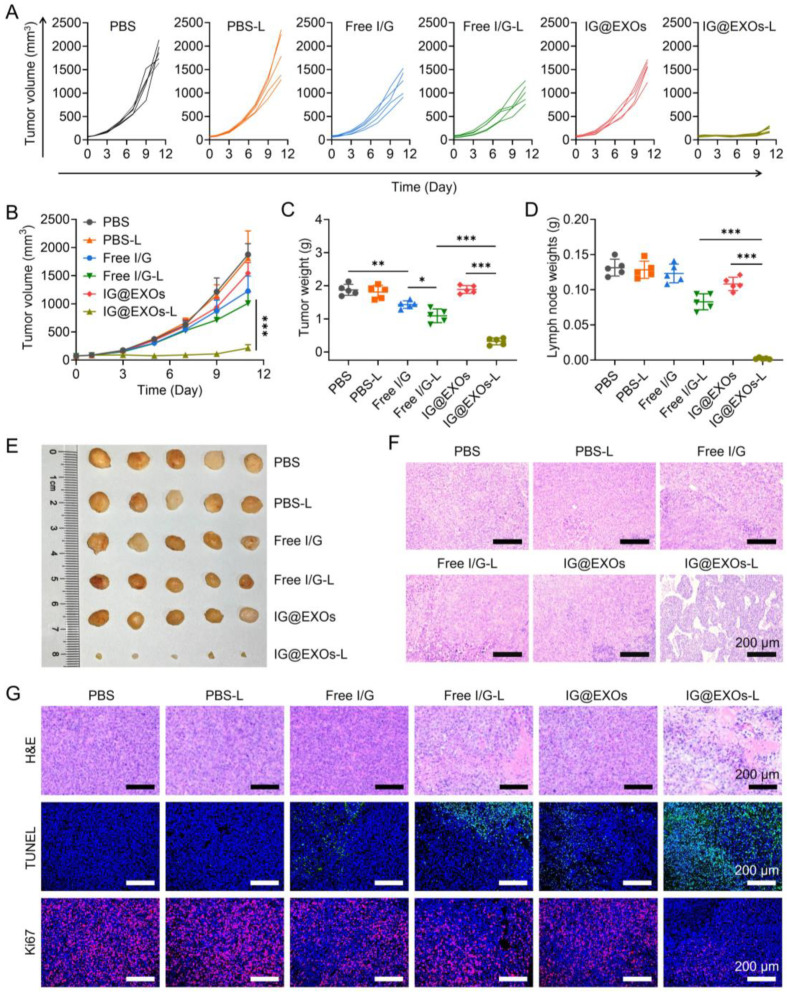
In vivo antitumor efficacy against SCC7 tumors. Spider plots of individual tumor growth curves (**A**), average tumor growth profiles (**B**), and average tumor weights (**C**) of mice bearing SCC7 subcutaneous tumors receiving various treatments. Weights (**D**), photograph (**E**), and H&E staining images (**F**) of the lymph nodes from mice treated with different formulations as indicated in (**A**). (**G**) H&E, TUNEL, and Ki67 analyses of tumor tissues from the SCC7 subcutaneous tumor-bearing mice receiving various treatments as indicated in (**A**). Statistical differences were defined as * *p* < 0.05, ** *p* < 0.01, and *** *p* < 0.001 in Figure 5D,F.

**Figure 7 pharmaceutics-16-00033-f007:**
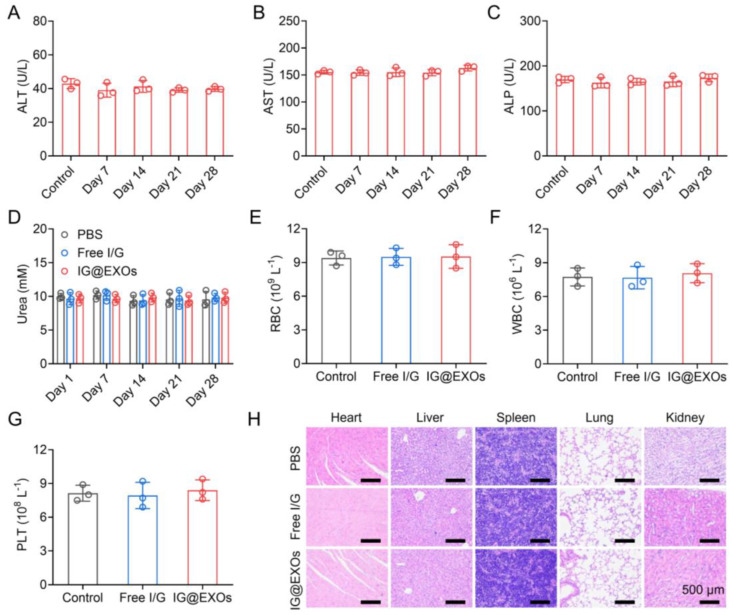
In vivo biosafety evaluation of IG@EXOs. Serum biochemistry assay of ALT (**A**), AST (**B**), ALP (**C**), and urea (**D**) of healthy mice after injection of IG@EXOs (7.5 mg kg^−1^, ICG) for 28 days. The number of RBC (**E**), WBC (**F**), and PLT (**G**) in the blood of mice treated with IG@EXOs or Free I/G at 3 days post-injection. (**H**) H&E analyses of the normal tissues, including the heart, liver, spleen, lung, and kidney, harvested from mice treated with IG@EXOs or Free I/G at 3 days post-injection.

## Data Availability

The data that support the findings of this study are available from the corresponding author upon reasonable request.

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
