# Peer review of "Synergistic Phototherapy-Molecular Targeted Therapy Combined with Tumor Exosome Nanoparticles for Oral Squamous Cell Carcinoma Treatment"

_pharmaceutics, 2023, doi:10.3390/pharmaceutics16010033_

Round 1

Reviewer 1 Report

Comments and Suggestions for Authors

Drug toxicity is a key problem in pharmacology, clinical medicine, and tumor therapies, and is one of the main reasons either for unpredictable unsuccess or high costs of drug development. Although this study has characterized the effects of the proposed method to deliver a well-known drug using a unique methodological approach, in truth there is always the risk of reaching erroneous conclusions. The action of the therapeutic strategy adopted in this case appears capable of inducing cell death in a very specific way, inhibiting the metabolism and/or blocking the proliferation of cancerous cells. However in this regard, the possibility of further affecting adjacent cells of healthy tissues is still unclear even in vitro... therefore on a possible future use, we aim to provide a note to researchers in the sector, advising them to also propose this possibility in vitro to improve the most suitable techniques for testing their compounds/hypotheses and drawing accurate and coherent conclusions.

Reviewer 2 Report

Comments and Suggestions for Authors

The authors investigate a nanoparticle system derived from tumor exosomes that is utilized to enhance the effectiveness of antitumor treatments against oral squamous cell carcinoma (OSCC). Their approach involves combining phototherapy and molecular targeted therapy to achieve a synergistic effect in combating the cancer. Tumor exosome-based nanoparticles may carry specific photosensitizer Indocyanine green (ICG) and EGFR inhibitor Gefitinib (IG@EXOs) that boosts the antitumor efficacy of OSCC.

I have a few suggestions for improvement of the paper:

1.       The title needs simplification. In the current form, the reader has difficulties to assimilate so vague written expression. One possibility to rewrite the title could be “Phototherapy combined with tumor exosome nanoparticles for Oral Squamous Cell Carcinoma treatment”.

2.       The authors may find suitable to point to a recent paper that report similar combinatorial therapy based on nanoparticles: “Dual and multi-drug delivery nanoparticles towards neuronal survival and synaptic repair”, Neural Regen Res. 2017; 12: 886–889; doi: 10.4103/1673-5374.208546

3.       The figures currently are grouped together after the body text of the paper. In order to avoid misplacement of the figure positions by the journal corrector, the authors could position them on the most suitable place in the text.

Reviewer 3 Report

Comments and Suggestions for Authors

Tumor exosome-based nanoparticle formulations were investigated for their ability to reduce oral squamous cell carcinoma.  The materials were fully investigated using common spectroscopy methods.  The nanoparticles were shown to penetration tumor cells and the mechanisms of actions were investigated.    The use of PMTT with IG@EXOs was capable of reducing tumors.  The authors suggest this is a new approach to reduce tumors.

The introduction is concise but lacks details.  The materials and methods section covers a great number of technical areas but lacks details and citations.  The results are clearly presented.  The figures are excellent.  The results are properly discussed.  The conclusions are supported by the results.  The introduction and materials/methods sections need improvement.

Specific comments to be addressed:

The introduction is very short, and does not cover the scope of the paper.  The total scope of the paper should be introduced in the introduction.

1)            The safety of the nanomaterials is not addressed in the introduction.

2)            The environmental impact of the materials investigated also is not covered. 

3)            The experimental section is well written.  However, certain sections lack details.

4)            Section 2.5 – what calculations were used for the photothermal conversion efficiency.

5)            Section 2.6 – why where the reactions stopped at 48 hours?

6)            In section 2.17. – the HPLC method should be cited so others can reproduce the work.

7)            Also in 2.17.  – the exact HPLC system should be stated, along with the parameters (mobile phase, column, etc). 

8)            The experimental details for the confocal laser scanning microscopy are missing.

9)            How was the tumor penetration monitored and assessed?

10)         The paper may benefit from a discussion of the merits and limitations of applying the technology in the real-world.

11)         A brief review and discussion of the safety of the materials beyond the results provided can be helpful.  This could include a discussion of the results with respect to published literature.
